# Effect of Professional and Extra-Professional Exposure on Seroprevalence of SARS-CoV-2 Infection among Healthcare Workers of the French Alps: A Multicentric Cross-Sectional Study

**DOI:** 10.3390/vaccines9080824

**Published:** 2021-07-27

**Authors:** Virginie Vitrat, Alexis Maillard, Alain Raybaud, Chloé Wackenheim, Bruno Chanzy, Sophie Nguyen, Amélie Valran, Alexie Bosch, Marion Noret, Tristan Delory

**Affiliations:** 1Infectious Diseases Department, Annecy Genevois Hospital, 74370 Epagny Metz-Tessy, France; avalran@ch-annecygenevois.fr; 2Clinical Research Unit, Annecy Genevois Hospital, 74370 Epagny Metz-Tessy, France; maillard.alexis@laposte.net (A.M.); mnoret@ch-annecygenevois.fr (M.N.); tdelory@ch-annecygenevois.fr (T.D.); 3Occupational Medicine Department, Alpes Leman Hospital, 74130 Contamine sur Arves, France; araybaud@ch-alpes-leman.fr; 4Infectious Diseases Department, Alpes Leman Hospital, 74130 Contamine sur Arves, France; cwackenheim@ch-alpes-leman.fr; 5Virology Laboratory, Annecy Genevois Hospital, 74370 Epagny Metz-Tessy, France; bchanzy@ch-annecygenevois.fr; 6Hygiene Unit, Annecy Genevois Hospital, 74370 Epagny Metz-Tessy, France; snguyen@ch-annecygenevois.fr; 7Infectious Diseases Department, Metropole Savoie Hospital, 73000 Chambery, France; alexie.bosch@ch-metropole-savoie.fr

**Keywords:** COVID-19, SARS-CoV-2, healthcare workers, cross-sectional survey, serologic testing

## Abstract

We aimed to report SARS-CoV-2 seroprevalence after the first wave of the pandemic among healthcare workers, and to explore factors associated with an increased infection rate. We conducted a multicentric cross-sectional survey from 27 June to 31 September 2020. For this survey, we enrolled 3454 voluntary healthcare workers across four participating hospitals, of which 83.4% were female, with a median age of 40.6 years old (31.8–50.3). We serologically screened the employees for SARS-CoV-2, estimated the prevalence of infection, and conducted binomial logistic regression with random effect on participating hospitals to investigate associations. We estimated the prevalence of SARS-CoV-2 infection at 5.0% (95 CI, 4.3%–5.8%). We found the lowest prevalence in health professional management support (4.3%) staff. Infections were more frequent in young professionals below 30 years old (aOR = 1.59, (95 CI, 1.06–2.37)), including paramedical students and residents (aOR = 3.38, (95 CI, 1.62–7.05)). In this group, SARS-CoV-2 prevalence was up 16.9%. The location of work and patient-facing role were not associated with increased infections. Employees reporting contacts with COVID-19 patients without adequate protective equipment had a higher rate of infection (aOR = 1.66, (95 CI, 1.12–2.44)). Aerosol-generating tasks were associated with a ~1.7-fold rate of infection, regardless of the uptake of FFP2. Those exposed to clusters of infected colleagues (aOR = 1.77, (95 CI, 1.24–2.53)) or intra-familial COVID-19 relatives (aOR = 2.09, (95 CI, 1.15–3.80)) also had a higher likelihood of infection. This report highlights that a sustained availability of personal protective equipment limits the SARS-CoV-2 infection rate to what is measured in the general population. It also pinpoints the need for dedicated hygiene training among young professionals, justifies the systematic eviction of infected personnel, and stresses the need for interventions to increase vaccination coverage among any healthcare workers.

## 1. Introduction

The global prevalence of SARS-CoV-2 infection was reported higher in healthcare workers than in the general population, at 8.7% (CI 95, 6.7% to 10.9%) [1,2]. It ranged from 0 to 45.3%, depending on the country, continent, and studies, and was around 8.5% in Europe [2].

In France, during the first wave, the national strategy was to detect symptomatic incident infections by reverse transcriptase polymerase chain reaction (RT-PCR). Yet, the prevalence of the infection, including asymptomatic forms, in French healthcare workers remains poorly known. In small-sized studies, it was ranging from 2.2% in serologic screening among asymptomatic healthcare workers to 28% in RT-PCR among symptomatic healthcare workers [3,4,5].

Demographics, patient-facing roles, and tests used for serologic testing can influence the estimation of infection prevalence [1,2]. In the general population, the uptake of personal protective equipment was shown to reduce virus transmission, but the effect of the type of occupation, protective equipment shortages, and hospital organization on the risk of infection is debated [2,6].

In this report, we aimed to estimate the seroprevalence of SARS-CoV-2 infection after the first wave of the pandemic among healthcare workers of the French Alps, and to explore if the infection rate varied by subgroups of care workers, types of occupation, hospital organization, or the uptake of protective equipment.

## 2. Materials and Methods

### 2.1. Study Context, Design, and Population

We conducted a multicentric cross-sectional study among four of the five public hospitals of the French Alps (NCT04845984). It was based on data issued from the mass serologic campaign initiated by the French Ministry of Health after the first wave of the pandemic, conducted at the national level among healthcare workers. The first wave of the pandemic ended at the time of the lockdown release on 11 May. According to national guidelines, from 27 June to 31 September 2020, any volunteer healthcare institution employees could reach the occupational medicine unit to be screened for SARS-CoV-2 infection by serology testing. Medical and nursing students were also invited to be screened. Before the serology testing, we invited the participants to fill out a self-questionnaire about (1) their demographics, (2) type and place of occupation, (3) being in a patient-facing role, (4) exposure to COVID-19 cases at work and in private life, (5) the use of personal protective equipment, and (6) symptoms of COVID-19. The self-questionnaire is available in Appendix A. We did not collect repeated data. At the beginning of the survey (27 June 2020), the daily incidence among the general population in the surveyed area was ranging from 0 to 0.2 per 100,000 inhabitants. At the end of the survey (31 September 2020), it increased to 8.3–32 per 100,000 inhabitants. The second national lockdown started on 29 October at a daily incidence of 163.5 to 203.7 per 100,000 inhabitants.

### 2.2. Serology Testing

The SARS-CoV-2 serology testing was done using the commercial test kits available in each participating hospital. Hospital 1 used the Abbott Architect SARS-CoV-2 IgG assay, Abbott Laboratories, Maidenhead, UK (IgG Sensitivity = 100%, IgG Specificity = 99.6%), the others used the Roche Elecsys Anti-SARS-CoV-2, Roche Diagnostics International Ltd, Rotkreuz, Switzerland (total antibody) assay (Pan IgG Sensitivity = 100%, Pan IgG Specificity = 99.8%). Sensibility and specificity are reported according to the EUA Authorized Serology Test Performance [7].

### 2.3. Healthcare Worker Classification

We classified healthcare workers according to the 2008 version of the International Standard Classification of Occupations (ISCO) [8]: (1) health professionals, (2) health associate professionals, (3) personal care workers in health services, (4) health management and support personnel, and (5) health service provider not elsewhere classified.

### 2.4. Sample Size

We enrolled 3454 healthcare workers, a sample that provided 84% power at a 5% bilateral first species risk (alpha) to detect a 4.4% (CI 95, 2.8% to 6.5%) prevalence of infection, as measured in the general population in the survey area after the first wave [9].

### 2.5. Statistical Analysis

We computed the frequencies and percentages for discrete variables and the median and interquartile range (IQR) for continuous variables. We used the chi-squared test to compare rates. We measured the prevalence of SARS-CoV-2 infection and its 95% confidence interval (CI 95) as the number of healthcare workers with positive serology testing over the number of tested personnel during the study. Because of the hospital effect, we used binomial logistic regressions with random effect (clustering) on the hospital to investigate the strength of associations between the presence of an infection in healthcare workers (outcome) and the variables included in the multivariable regression. The unit of analysis was the individual. Variables included in the multivariable regression were defined a priori, and no automatic variable selection was performed: age, sex, healthcare worker occupation according to the ISCO, student status, patient-facing role, professional exposure to SARS-CoV-2 (working in a COVID-19 unit, performing an aerosol-generating task, cases among colleagues), uptake of personal protective equipment, working in an emergency ward, working remotely full-time, and contact with intra- and extra-familial cases. Associations are reported as the odds ratio (OR) and their CI 95. All tests were two-tailed, and the level of significance was set at 5% bilateral. Analyses were performed on R, version 4.0.1 (R Foundation for Statistical Computing, Vienna, Austria), using the ‘glmmML’ and ‘ggplot2′ packages.

### 2.6. Patient and Public Involvement

No patients were involved in setting the research questions or the outcome measures, nor were they involved in developing plans for the study design. No patients were asked for advice on the interpretation or the writing up of the results.

## 3. Results

### 3.1. Characteristics of Respondents

We enrolled 3454 staff members across four hospitals, corresponding to a 28.3% participation rate, and to 77.6% of professionals who underwent serology testing (Figure 1). In Hospital 3, the serologic screening was only proposed in a few wards, and 8.5% of the personnel were tested for serology. In others, 50.4% to 62.5% of healthcare workers were screened. According to the ISCO, 1818 (52.6%) of staff members were healthcare professionals, 766 (22.2%) were health-associated professionals, 854 (24.7%) were health management and support personals, and 16 (0.5%) did not report their occupation. Workers had a median age of 40.6 years old (31.8–50.3), 83.4% were female, and 631 had performed RT-PCR before the study, of which 13.3% were positive. Table 1 details the staff members’ characteristics overall and by hospital.

Many participants reported symptoms compatible with COVID-19 infection. The presence of symptoms by serological status is presented on Appendix B.

### 3.2. Local Prevention Measures and Epidemiology in Each Hospital

The medico-surgical bed capacity ranged from 265 beds in Hospital 4 to 972 beds in Hospital 3. All but Hospital 4 had 12 to 18 ICU beds before the pandemic. Infection control teams and infectious disease specialists were available in all the participating hospitals before the pandemic. During the first wave, 1319 COVID-19 cases were admitted to the participating hospitals, ranging from 249 (Hospital 4) to 533 (Hospital 1). At the peak, COVID-19 cases were occupying 9.5% to 21.9% of the medico-surgical beds. Patients with COVID-19 were mostly hospitalized in dedicated units, with a ratio of 1 nurse per 7 to 10 beds. All hospitals had increased ICU bed capacity beyond 200%. All-cause mortality among COVID-19 cases ranged from 6.5% to 14.9%. Every hospital recommended systematic surgical mask-wearing, regardless of the patients’ COVID-19 status. It was first supported for professionals with a patient-facing role, then extended to any healthcare worker. In Hospitals 1 and 3, it was even recommended before the pandemic, as part of the usual hospital-acquired flu prevention strategy. Because of shortages, Hospitals 1 and 2 increased the using time of surgical masks to 8 h instead of 4 h. None of the hospitals faced shortages of disposable filter respiratory protection (N95/FFP2). Appendix C details the characteristics of the participating sites.

### 3.3. Prevalence of SARS-CoV-2 Infection

The overall seroprevalence of SARS-CoV-2 infection was 5.0% (CI 95, 4.3% to 5.8%). It varied by hospital, ranging from 3.3% (CI 95, 2.5% to 4.1%) to 10.4% (CI 95, 7.7% to 13.6%).

### 3.4. Factors Associated with an Increased Prevalence

Figure 2 shows the seroprevalence of infection by type of occupation according to the ISCO. The type of occupation, according to the ISCO, was not associated with higher prevalence.

Figure 3 shows the seroprevalence of infection according to healthcare workers’ characteristics. Young professionals below 30 years old (aOR = 1.59, (CI 95, 1.06 to 2.37)), including paramedical students and residents (aOR = 3.38, (CI 95, 1.62 to 7.05)), showed an increased rate of infection. The location of work, including emergency wards and COVID-19 units, was not associated with an increase in infections.

Table 2 summarizes the results of univariable and multivariable associations with seroprevalent infections. Staff members in patient-facing roles did not show an increased likelihood of infection. Healthcare workers reporting contact with COVID-19 patients without adequate protective equipment had a higher infection rate (aOR = 1.66, (CI 95, 1.12 to 2.44)). However, the systematic wearing of surgical face masks was not associated with decreased seroprevalence. For those who performed aerosol-generating procedures, the use of an N95/FFP2 mask did not reduce the rate of infection. Note that the wearing of an N95/FFP2 mask was not recommended for aerosol-generating tasks in patients without confirmed COVID-19. In addition, generalized screening of SARS-CoV-2 infection among any newly admitted patients was not performed during the study period. Healthcare workers exposed to clusters of COVID-19-infected colleagues (aOR = 1.77, (CI 95, 1.24 to 2.53)) or intra-familial COVID-19 relatives (aOR = 2.09, (CI 95, 1.15 to 3.80)) had a higher likelihood of infection, whereas healthcare workers with extra-familial exposure to COVID-19 cases did not show an increased rate of infection.

## 4. Discussion

The seroprevalence of SARS-CoV-2 infection among healthcare workers of the French Alps was 5.0% (CI 95, 4.3% to 5.8%) after the first wave of the pandemic. It was higher in young professionals (including students), in those performing an aerosol-generating procedure, those exposed to COVID-19 cases without adequate uptake of protective equipment, and those reporting contact with clusters of infected colleagues and intra-familial cases.

We showed a relatively low prevalence of infection (5.0%) after the first wave. This is in the 3.4% to 11.2% range described in other European surveys [10,11,12,13,14]. We estimated that the prevalence (inverse variance method with random effect) among French healthcare workers was 12.0% (CI 95, 7.0% to 19.0%), varying by the sampling method, and ranging from 7.0% when considering any healthcare workers to 12% among frontline caregivers, and 31.0% among symptomatic health professionals (Appendix D) [3,5,15,16,17,18]. Such a variation in estimation highlights the need to identify subgroups at higher risk of infection. In our multi-centric study, we estimated the effect of individual uptake of protective equipment and intra-/extra-professional exposure to the virus on the infection rate.

We found the lowest prevalence in health professional management support (4.3%), which served as a proxy for the general population. Over the same timeframe, Le Vu et al. reported a 4.4% prevalence within the surveyed area’s general population [9]. Other studies compared the prevalence among healthcare workers to the general population without a proper internal control group. They showed a higher likelihood of infection among healthcare workers but did not report on the availability and type of protective equipment used, its uptake by healthcare workers, nor organizational characteristics at the hospital level [10,19,20]. In our study, none of the participating hospitals faced a real shortage of protective equipment. The organizational and epidemiological characteristics of the hospitals were fairly homogeneous, as were the implementation of local guidelines for the systematic wearing of a facemask by healthcare workers.

Some studies have reported that males are at a higher risk of infection than females [13,17]. Females represented 83.4% of our sample and this may have contributed to a lower seroprevalence. However, the administrative data describing the workforce of the participating sites showed the same sex ratio. This low prevalence is likely to reflect a low circulation of the virus within the surveyed area, rather than a biased selection towards women. Only a third of professionals underwent serology testing, of which we enrolled three-quarters. A sampling bias can thereby exist for professionals previously infected, resulting in an underestimation of prevalence.

In our study, professionals who performed aerosol-generating procedures and who were exposed to COVID-19 patients without appropriate protective equipment were at higher risk of infection, unlike those facing patients (infected or not) or working in a COVID-19 unit. This contrasts with previous reports [1,10,14,20,21,22,23]. This suggests that a lack of compliance with hygiene measures drives the risk of infection rather than the location of practice. Indeed, at the time of the study, N95/FFP2s were used for aerosol-generating tasks in COVID-19 patients only (oral intubation, aerosolized therapy, high-flow oxygen, etc.). A French survey among healthcare workers showed a lower use of protective equipment in non-COVID-19 units (0% to 51%) than in high-risk areas (56% to 87%) [24]. These results stressed the implementation of reinforced preventive measures in our hospitals, including the systematic use of N95/FFP2 for any aerosol-generating task. Since November 2020, a generalized screening of COVID-19 infection by RT-PCR at hospital admission has also been implemented.

The professional category was not associated with an increased prevalence of infection, but young professionals (8.3%), including paramedic students and residents, had a higher infection rate (16.9%) compared to others (4.0% to 4.7%). A Danish study found similar results because of a hotspot identified among medical students attending a social gathering at a university club [25]. Hygiene training dedicated to students should be promoted.

Finally, in our study, the strongest associations with SARS-CoV-2 infection were not related to close contact with patients. Indeed, students and workers in contact with COVID-19 cases among their colleagues or relatives had the highest infection rate. In the literature, few articles have highlighted that cross-transmission between healthcare workers can occur [11,14]. During the first wave, most infected health professionals continued working unless being severely symptomatic. On February 16, 2021, the French Ministry of Health decided to ban any infected professional from work. Our results thereby support the eviction of infected personnel from the hospital to prevent cross-transmission between staff members. Intra-familial exposure was already shown to drive infections, but little can be done to limit this risk [14,22].

Our study was limited by the six-month lag between the first wave peak and the end of serological sampling. Some studies showed an antibody titer decrease three to six months after infection in more than half of the infected people [26]. Despite the high performance of serological kits used, this may have contributed to a low seroprevalence. Second, inclusions were not exhaustive, and sampling bias could have occurred. Third, the cross-sectional design did not allow for collecting repeated data or for drafting causalities. Nevertheless, we investigated the uptake of protective equipment and hygiene measures at an individual level. Even if measurement biases are likely with our design, the lack of shortage in protective equipment over the studied period reinforces the confidence in the interpretation of associations. We also used health management support professionals as an internal control group, allowing us to compare our results with the general population.

## 5. Conclusions

In a general population of healthcare workers with sustained availability of personal protective equipment, the rate of SARS-CoV-2 infection was low and comparable to that of the general population. Young professionals are particularly at risk and may benefit from dedicated hygiene training. Cross-transmission between healthcare workers is a real threat to care continuation. It justifies the systematic eviction of infected personnel and stresses the need for interventions to increase vaccination coverage among any healthcare workers.

## Figures and Tables

**Figure 1 vaccines-09-00824-f001:**
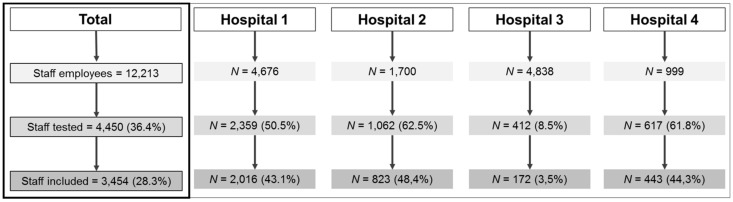
Flowchart of participant enrolment.

**Figure 2 vaccines-09-00824-f002:**
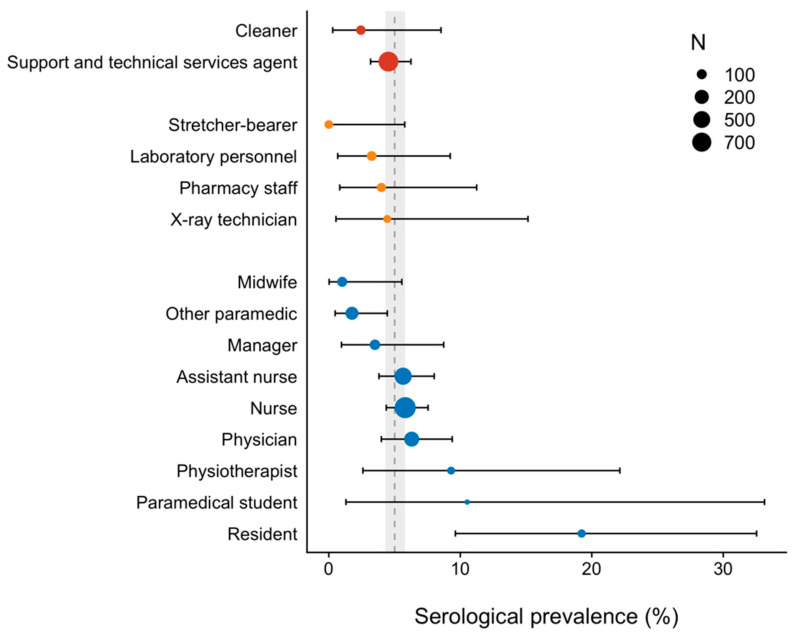
Seroprevalence of SARS-CoV-2 infection by type of occupation according to the ISCO. Health management and support personnel are presented in red, health associate professionals are yellow, and health professionals are blue. The size of the dots represents the number of subjects. The vertical dashed line is the estimated seroprevalence in the healthcare worker population with its 95% confidence interval (grey area).

**Figure 3 vaccines-09-00824-f003:**
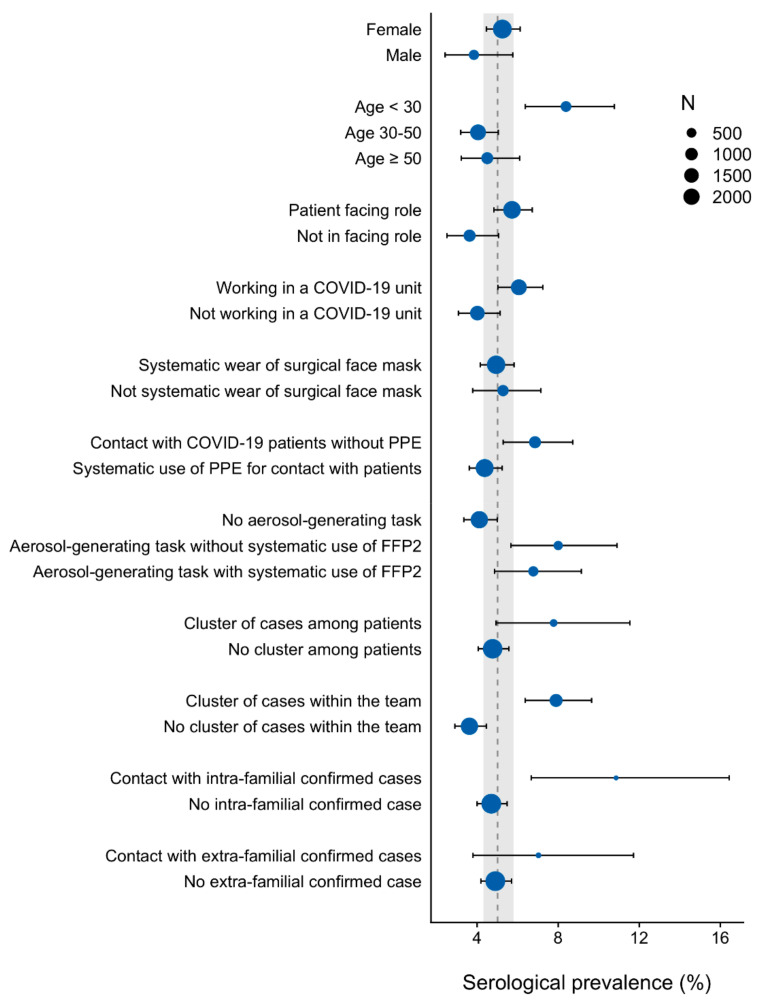
Seroprevalence of SARS-CoV-2 infection by characteristics of healthcare workers. The size of the dots represents the number of subjects. The vertical dashed line is the estimated seroprevalence in the healthcare worker population with its 95% confidence interval (grey area).

**Table 1 vaccines-09-00824-t001:** Characteristics of healthcare workers, overall and by participating hospital.

Healthcare Worker Characteristics	Missing (%)	Overall	Hospital 1	Hospital 2	Hospital 3	Hospital 4
		*N* = 3454	*N* = 2016	*N* = 823	*N* = 172	*N* = 443
*Demographics*		*N (%) or Med (IQR)*	*N (%) or Med (IQR)*	*N (%) or Med (IQR)*	*N (%) or Med (IQR)*	*N (%) or Med (IQR)*
Age, continuous (years)	2.2%	40.6 (31.8, 50.3)	40.2 (31.8, 49.2)	41.2 (32.0, 51.2)	40.3 (32.6, 50.9)	41.8 (31.3, 51.4)
Sex, female	0.0%	2880 (83.4)	1667 (82.7)	683 (83.1)	145 (84.3)	385 (86.9)
BMI, continuous (kg/m2)	6.0%	22.6 (20.4, 25.1)	22.3 (20.3, 24.7)	23.2 (20.8, 26.0)	22.7 (20.8, 24.7)	22.5 (20.2, 25.2)
Children at home	10.2%	1695 (54.6)	1031 (56.6)	396 (53.7)	80 (51.0)	188 (48.3)
Household, >1 inhabitant	0.0%	2078 (60.2)	1232 (61.1)	499 (60.6)	100 (58.1)	247 (55.8)
***Occupation according to ISCO †***	0.5%					
Health management and support personnel		854 (24.7)	482 (23.9)	225 (27.3)	30 (17.4)	117 (26.4)
Health associate professionals		766 (22.2)	442 (21.9)	170 (20.7)	49 (28.5)	105 (23.7)
Health professionals		1818 (52.6)	1080 (53.6)	425 (51.6)	93 (54.1)	220 (49.7)
Missing data		16 (0.5)	12 (0.6)	3 (0.4)	0 (0.0)	1 (0.2)
***Extra-professional exposure***						
Intra-familial confirmed cases	0.0%	175 (5.1)	96 (4.8)	41 (5.0)	11 (6.4)	27 (6.1)
Extra-familial confirmed cases	0.1%	185 (5.4)	107 (5.3)	39 (4.8)	18 (10.5)	21 (4.7)
***Professional exposure***						
Working in emergency ward	0.1%	282 (8.2)	151 (7.5)	57 (6.9)	0 (0.0)	74 (16.7)
Working in COVID-19 unit	2.7%					
Never		1496 (44.5)	885 (45.3)	380 (47.4)	81 (47.1)	150 (34.6)
Sometimes		749 (22.3)	465 (23.8)	175 (21.8)	12 (7.0)	97 (22.4)
Often		382 (11.4)	209 (10.7)	90 (11.2)	12 (7.0)	71 (16.4)
Always		734 (21.8)	396 (20.3)	156 (19.5)	67 (39.0)	115 (26.6)
Working in COVID-19 intensive care unit	2.8%	134 (4.0)	71 (3.6)	30 (3.8)	27 (15.7)	6 (1.4)
Working in COVID-19 room	2.8%	1730 (51.5)	999 (51.1)	390 (48.8)	64 (37.2)	277 (64.0)
Contact with COVID-19 patients without PPE ‡	3.9%	866 (26.1)	458 (23.9)	200 (25.2)	30 (17.4)	178 (41.3)
Cluster of cases among the team	0.1%	1102 (31.9)	578 (28.7)	245 (29.8)	67 (39.0)	212 (48.1)
Cluster of cases among patients	0.0%	283 (8.2)	188 (9.3)	50 (6.1)	6 (3.5)	39 (8.8)
Performed aerosol-generating tasks	2.4%	1033 (30.7)	616 (31.5)	221 (27.5)	70 (41.2)	126 (28.6)
N95/FFP2 mask during aerosol generating-tasks	2.6%					
Not applicable		2337 (69.5)	1341 (68.6)	582 (72.8)	100 (58.8)	314 (71.7)
Never		82 (2.4)	54 (2.8)	21 (2.6)	2 (1.2)	5 (1.1)
Sometimes		107 (3.2)	59 (3.0)	32 (4.0)	2 (1.2)	14 (3.2)
Often		261 (7.8)	158 (8.1)	61 (7.6)	11 (6.5)	31 (7.1)
Always		576 (17.1)	343 (17.5)	104 (13.0)	55 (32.4)	74 (16.9)
Systematic wear of surgical face mask	0.0%	2715 (78.6)	1601 (79.4)	612 (74.4)	163 (94.8)	339 (76.5)
***SARS-CoV-2 infection***						
Seroprevalence of COVID-19	0.0%	173 (5.0)	66 (3.3)	53 (6.4)	8 (4.7)	46 (10.4)
Reporting symptoms compatible with COVID-19	0.0%	2254 (65.3)	1346 (66.8)	493 (59.9)	98 (57.0)	317 (71.6)
Time between symptoms and serological screening	34.8%					
≤14 days		104 (4.6)	61 (4.5)	20 (4.1)	13 (13.4)	10 (3.2)
15–29 days		194 (8.6)	112 (8.3)	36 (7.3)	21 (21.6)	25 (7.9)
≥30 days		1857 (82.5)	1102 (82.0)	423 (85.8)	62 (63.9)	270 (85.2)
Not disclosed		96 (4.3)	69 (5.1)	14 (2.8)	1 (1.0)	12 (3.8)
Performed RT-PCR for COVID-19	0.1%	631 (18.3)	263 (13.1)	196 (23.8)	77 (44.8)	95 (21.4)
Positive		84 (13.3)	35 (13.3)	22 (11.2)	14 (18.1)	13 (13.7)
Negative		547 (86.7)	228 (86.7)	174 (88.8)	63 (81.8)	82 (86.3)

† ISCO: International Standard Classification of Occupation. ‡ PPE: Personal protective equipment.

**Table 2 vaccines-09-00824-t002:** Univariable and multivariable binomial logistic regression associations with SARS-CoV-2 infection among healthcare workers (N = 3299).

Variables	Univariate		Multivariate			
	Odds Ratio	CI 95	*p*-Value	Adjusted Odds Ratio	CI 95	*p*-Value
Age (*ref. 30 to 50 years*)						
<30 years	2.17	(1.52–3.11)	<0.001	1.59	(1.06–2.37)	0.024
>50 years	1.12	(0.75–1.66)	0.581	1.28	(0.83–1.96)	0.259
Female (*ref. male*)	1.39	(0.88–2.19)	0.161	1.55	(0.94–2.54)	0.085
Profession, according to ISCO † *(ref. health management and support personnel)*						
Health associate professionals	1.12	(0.70–1.79)	0.632	0.87	(0.50–1.53)	0.634
Health professionals	1.27	(0.86–1.87)	0.223	0.67	(0.39–1.16)	0.157
Paramedical student or resident *(ref. not a student)*	4.05	(2.13–7.69)	<0.001	3.38	(1.62–7.05)	0.001
Patient-facing role *(ref. not in facing role)*	1.61	(1.09–2.38)	0.016	1.10	(0.65–1.87)	0.712
Working in emergency ward *(ref. not in emergency ward)*	1.51	(0.93–2.44)	0.097	0.85	(0.49–1.49)	0.571
Working in a COVID-19 unit *(ref. not in COVID-19 unit)*	1.54	(1.12–2.13)	0.008	1.03	(0.67–1.58)	0.899
Contact with COVID-19 patients without PPE ‡ *(ref. systematic use of PPE for contact with patients)*	2.37	(1.73–3.25)	<0.001	1.66	(1.12–2.44)	0.011
Systematic wear of surgical face mask *(ref. not systematic)*	0.93	(0.65–1.34)	0.706	0.71	(0.45–1.13)	0.151
Aerosol-generating task *(ref. not concerned)*						
Systematic use of FFP2	1.70	(1.15–2.49)	0.007	1.74	(1.06–2.85)	0.028
Without systematic use of FFP2	2.03	(1.36–3.02)	<0.001	1.81	(1.09–3.01)	0.021
Cluster of cases among patients *(ref. not)*	1.69	(1.06–2.68)	0.028	1.31	(0.79–2.19)	0.299
Cluster of cases within the team *(ref. not)*	2.28	(1.68–3.11)	<0.001	1.77	(1.24–2.53)	0.002
Working remotely at full-time *(ref. not)*	1.41	(0.51–3.95)	0.509	2.39	(0.79–7.19)	0.121
Intra-familial confirmed cases *(ref. not)*	2.47	(1.49–4.09)	<0.001	2.09	(1.15–3.80)	0.015
Extra-familial confirmed cases *(ref. not)*	1.47	(0.82–2.64)	0.199	0.81	(0.40–1.64)	0.555

† ISCO: International Standard Classification of Occupation. ‡ PPE: Personal protective equipment.

## Data Availability

The authors confirm that the data supporting the findings of this study are available within the article (and/or) the Appendix A, Appendix B, Appendix C, and Appendix D.

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
