# Peer review of "Effect of Professional and Extra-Professional Exposure on Seroprevalence of SARS-CoV-2 Infection among Healthcare Workers of the French Alps: A Multicentric Cross-Sectional Study"

_vaccines, 2021, doi:10.3390/vaccines9080824_

Round 1

Reviewer 1 Report

Manuscript Title: Effect of professional and extra-professional exposure on sero-2 prevalence of sars-cov-2 infection among healthcare workers of 3 the french alps: a multicentric cross-sectional study.

General comments:

In this study the authors have performed a multi-centric cross-sectional study to determine the seroprevalence of SARS-CoV2 infection among healthcare workers of French alps. The authors have used a large data set of 3454 healthcare workers and found that the prevalence of SARS-CoV-2 infection was around 5%. They have also found that the infection is more frequent in young professionals below 30 years of age. The authors have also checked other factors like location of work and found no association with increased infection. The authors conclude that the in order to limit the cross-transmission of the virus from the patients to healthcare workers proper and efficient practice at individual levels. Although, the study is well done with enough data sets, it seriously lacks any novelty at this stage. It is already well known that personal protective gears and individual uptake helps in decreasing the virus transmission. Moreover, a large number of similar studies have already been done in other populations, therefore this study does not add any extra or unknown phenomenon. Apart from that there are many typos in the manuscript that needs to be taken care of. Due to the above-mentioned reasons, I therefore recommend against publishing this article in this journal in the current form.

Author Response

We thank the reviewer for his remarks.

Within the last year, prevalence of SARS-COV-2 infection has been the subject of many studies and publication. Nevertheless, the risk evaluation remains somehow unclear for highly exposed healthcare workers who appropriately used personal protective equipment.

The literature is discordant regarding the risk stratification of Covid 19 infection rate by the type of occupation. The main occupational risk factors reported appears to be direct exposure to infected patients for professionals in a facing role (Shaah et al). However, previous studies, did not detail the availability and uptake of individual protective equipment, which may have been lacking during the first wave. The assessment of risk factors is therefore incomplete. In our opinion, the main strengths of our study are:

- The investigation association between SARS-CoV-2 infection with professional and extra-professional factors at an individual level, among caregivers benefiting from readily available and suitable protection.

- The measurement of hospital organization.

We modified the manuscript to highlight these points.

We also corrected the typos.

Reviewer 2 Report

In this manuscript entitled “Effect of professional and extra-professional exposure on sero- prevalence of sars-cov-2 infection among healthcare workers of the french alps: a multicentric cross-sectional study” the authors have explored the seroprevalence after the first wave of the covid19 pandemic among healthcare workers, to better understand the infection rate in this specific population (that is intrinsically more exposed) and identify protective patterns.

Although the conclusions are not especially novel, this article is still important mainly due to its large sample size and to put specific numbers on some issues that should be in mind of politicians and the general population alike.

Indeed, this article highlights the importance of giving our healthcare workers the protective equipment and measures to perform their work safely.

The manuscript is fairly interesting, well designed and appropriately analyzed, but there are many caveats that hinders its publication. Further limitations are given in the specific comments.

Major comments

  • The authors state in the introduction that “the effect of the type of occupations, the level of exposure to SARS-CoV-2, the use and uptake of personal protective equipment, and hospital organization on infection and virus spread remains unclear”. This is only partially true; the level of exposure to SARS-CoV-2 and the use and uptake of personal protective equipment is already well established, and should not be mentioned here. However, the effect of type of occupation and hospital organization as independent variables in risk of infection is clearly novel and should be the main focus of the article.
  • Most of the data presented is of common knowledge already (that PPE works, young people is most exposed, not-protected encounters give more room for higher infection rates…). The authors should try and focus on what is really novel from this study. For example: A more developed granular data to better understand the specific risks of each subgroup of professionals or to better highlight the social impact on how the students (residents especially) have taken a serious toll on the front-lines.
  • In the subsection 3.3 of the results (Page 6 line 149), the authors should clearly explain the main points that the reader should focus on the results, not only attach a table.
  • The authors should undergo a light revision of the English grammar.

Minor comments

  • Why do the authors publish this article now? The data was collected until September and the analyses should not require that much time.
  • As of June 27, the daily incidence in the surveyed area's general population was ranging from 0 to 0.2 per 100 000-inhabitants, while it was 8.3 to 32 per 100 000-inhabitants by the end of the survey”. Please, clarify the dates to make it easier to visualize with specific time frames.
  • The authors appropriately explain that 1200 subjects would provide a 75% precision with 5% alpha, which is the standard minimum. It would be appropriate to specify then the precision obtained with their 3454 sample size, or at least indicate that this is their final subject population in the “sample size” subheading.
  • In the subsection 3.2 of the results (Page 5 line 134): “During the first wave, 249 (Hospital 4) to 533 (Hospital 1) COVID-19 patients were admitted”. How many patients in total (all 4 hospitals)? 
  • In the same sub section, page 5 line 140: “It was first supported for professional in a facing role, then extended to any healthcare worker. In hospital 1 and 3, it was even recommended before the pandemic, as part of the usual hospital-acquired flu prevention strategy.” How do the authors account for the disparity between professions adopting the use of PPE? Especially in the beginnings of the pandemic, where the strict norms were not enforced fully, each day of work or contact could have had a great impact on the transmission of the virus.
  • Table 2 should be supplementary, because it is important to report this data, but at this time the symptoms of COVID-19 are well known and there seems to be nothing especial in this subpopulation that merits highlight.

Reviewer 3 Report

The main aim of this study is to report the seroprevalence of SARS-CoV-2 infection after the first wave of the pandemic among healthcare workers of the French Alps, and to explore the factors associated with increased infection rate. The paper can be accepted for publication after taking into consideration the following comments:

  1. Give more graphic representations in order to illustrate your results.
  2.  The choice of statistical methods should be more justified.
  3.  The comparison of your results with other works existing in the literature should be added.
  4. Conclusion section needs to be improved.
  5. There are some typos. The authors should carefully read the manuscript.
  6.  Check and unify the citations of references.

Round 2

Reviewer 2 Report

The authors have appropriately responded to all the comments and made the necessary changes to clarify the manuscript.

Reviewer 3 Report

Now, the paper is well written and well organized.  Therefore, I recommend it for the publication in the journal.